# Dynamic pricing optimization for high-speed railway based on passenger flow assignment

**Jiren CAO**[1,2], **Lei NIE**[1,2], **Lu TONG**[1,2]*, **Zhenhuan HE**[1,2], **Zhangjiaxuan LIU**[3]

**1** Frontiers Science Center for Smart High-speed Railway System, Beijing Jiaotong University, Beijing, China, **2** School of Traffic and Transportation, Beijing Jiaotong University, Beijing, China, **3** Institute of Computing Technologies, China Academy of Railway Sciences Corporation Limited, Beijing, China

* 15251066@bjtu.edu.cn

**Data Availability Statement:** All relevant data are within the manuscript and its Supporting information files.

**Funding:** This research was supported by High-Speed Rail Joint Funds of the National Natural

## Abstract

In order to improve the operation efficiency and market competitiveness, how to optimize the ticket pricing strategy of high-speed railway to match the dynamic supply-demand relationship was an urgent problem to be studied. Taking differentiated passenger demand and supply trains as the research object, the space-time service network based on train timetable was constructed. The generalized cost formula and travel utility formula of passenger travel were proposed, which contained economy, rapidity, convenience, comfort, and route correlation cost. A multi-objective dynamic pricing model was proposed. The model aimed at maximize the corporate revenue and maximize passenger travel benefit, and was solved by large neighborhood search heuristic algorithm and path size logit assignment based on capacity constraint-passenger flow increment accurate algorithm. Based on real data, the *Shandong circular high-speed railway* case compared the average total revenue under different ticket price adjustment ranges and the ticket price for different classes of trains under different OD levels. The case proved the practicability of dynamic pricing adjustment strategy considering train classification, which could provide a reference for the ticket price management of high-speed railway.

## 1 Introduction

Large-scale and complex high-speed railway network provides a strong capacity support for the marketization of railway transport. The design quality of passenger transport products has a direct impact on the operational efficiency, service quality and market competitiveness of high-speed railway network. Dynamically adjusting the balance between supply and demand has become a key problem to be solved urgently in the development of high-speed railway, and dynamic pricing is one of the important means.

The dynamic pricing mechanism of China high-speed railway is still in the initial stage, which urgently needs systematic scientific research. Compared with air and road transportation, the overall process of railway marketization is slow, and the ticket prices of each line remain relatively fixed, which has not yet formed a scientific and effective adjustment method.

Science Foundation of China (Grant No.
U2368212); Fundamental Research Funds for the
Central Universities (Science and technology
leading talent team project, No. 2022JBQY005);
Young Scientists Fund of National Natural Science
Foundation of China (72001021);The research plan
of China Railway(K2023X047;K2023X031;
P2023X029).

**Competing interests:** The authors have declared
that no competing interests exist.

In the process of high-speed railway market-oriented operation, it has become an important development trend of China high-speed railway operation to formulate a scientific dynamic decision-making mechanism for ticket prices and realize the effective assignment of transportation resources to balance the relationship between market supply and demand.

Outside the high-speed railway market, the dynamic pricing strategy can improve the market competitiveness of high-speed railway products. In the internal high-speed railway market, the dynamic pricing strategy can adjust the arrival and departure time of passenger flow, which has huge economic benefits and extensive social benefits.

From the perspective of scientific research, the dynamic pricing outside the pre-sale period is the basis of the dynamic pricing during the pre-sale period. From the perspective of market-oriented reform, the initial stage of reform needs to be steady. Therefore, the scenario is set to the dynamic pricing outside the pre-sale period.

For the dynamic pricing scenario without pre-sale period, the dynamic pricing problem of multiple trains with multiple ODs in the space-time service network is described in this study. Aiming to solve the problem of how the dynamic pricing of high-speed railway can not only meet the travel demand of passengers, but also increase the total revenue, a research idea is put forward based on differentiated train products, that dynamic pricing is the main factor and passenger flow assignment is the auxiliary factor.

This paper is organized as follows: Section 2 gives the literature review, research ideas and innovation points; Section 3 is the definitions and locations of key notations and parameters; Section 4 describes the basic concepts of passenger flow assignment and the relevant formulas of passenger choice behavior; Section 5 introduces the multi-objective model and the comprehensive algorithm, including the heuristic algorithm of large-scale neighborhood search and the accurate algorithm of path size logit passenger flow assignment; Section 6 is the case study of *Shandong circular high-speed railway* based on real timetable data and passenger flow data; Section 6 presents the conclusions.

## 2 Literature review

The pricing of high-speed railway passenger transport products is an important guarantee to enhance the competitiveness of passenger transport market and transportation income. High-speed railway passenger flow assignment is the key technology to simulate passenger flow choice behavior under network.

The pricing optimization of high-speed railway is a dynamic pricing problem in revenue management. The dynamic pricing strategies are mainly applied to air transportation, hotel management, automobile service and railway transportation. [1–3] focused on the revenue management of perishables products, which include airline seats, railway seats, hotel rooms, theatre seats, seasonal goods, etc.

The existing researches can be roughly divided into two parts: pure dynamic pricing and ticket price cooperative optimization, which involve the relevant content of passenger choice behavior. This paper further extends from passenger choice behavior to passenger flow assignment. Therefore, the literature review will be described in the following three parts.

In terms of pure dynamic pricing of high-speed railway [4], suggested that the railway department should make ticket price according to the peak period and the off-peak period of passenger flow. [5] proposed the ticket price should be set in lower level on weekdays and higher level on holidays. [6] proposed a dynamic pricing model of high-speed railway from the perspective of groups and individuals. [7] used epsilon constraint method to study the pricing of high-speed railway. [8] studied the impact of airline price discrimination on the profits and social welfare of high-speed railway and airline.

In term of passenger choice behavior and passenger flow assignment [9], pointed out that the path-overlap problem occurred when Multinomial Logit (*MNL*) faced the road network, which violated the IIA independent assumption. The paper deduced three formulas of Path Size Logit (*PSL*), and pointed out that the original formula should be preferred because of its theoretical basis. [10] presented an estimation method and algorithm based on long-term transaction data for railway passengers. [11] presented a train stop scheduling method based on passenger flow assignment. [12] analyzed the choice behavior preferences of high-speed railway passengers. The case results showed that with the increase of travel distance, passengers' sensitivity to ticket price and speed increased correspondingly. [13] presented an integrated Bayesian statistical inference framework to characterize passenger flow assignment models in a complex subway network. [14] used disaggregate model to study the choice behavior for traveler in comprehensive transportation corridor and found the advantage distance of high speed railway was about 500–1000 km. By adjusting ticket prices, the market shares of different mode could be effectively optimized. [15] proposed a flexible passenger priority assignment framework for public transport. Aiming at maximizing passenger flow. [16] combined passenger flow assignment with ticket, but did not involve fare related content. [17] developed a user equilibrium dynamic traffic assignment model to capture path choice behavior in dockless bike sharing systems. Although the dynamic pricing scheme was involved, but more of it belongs to the problem of passenger flow assignment. [18] proposed a route selection model considering route correlation for subway network scenarios. [19] studied the heterogeneous behavior of metro passengers under unplanned service disruption with uncertainty. [20] captured passenger behavioral responses to the passenger flow control strategies through stated preference-off-revealed preference survey. [21] presented a simplified bi-layer network framework to find the K-shortest path as early as possible to improve the speed of flow assignment while meeting accuracy requirements. [22] analyzed the decision-making behavior of Chinese central city passengers on high-speed railway and air from the perspective of socioeconomic level and fare. [23] established the conditional logit model and analyzed the ticket purchase influencing factors of business passengers and non-business passengers. Aiming at the integrated mode of cross-line and skip-stop operation between state and suburban railway [24], proposed a passenger flow assignment model based on the path-sized logit and designed an improved method of successive averages algorithm. [25] simulated vehicle operation and passenger travel in urban rail transit network by using connected vehicles and unconnected passengers.

In terms of ticket price cooperative optimization of high-speed railway [26], incorporated multiple logit model and latent class model into the revenue optimization problem, the case results showed that accepting short-haul demand could provide greater revenue than long-haul demand using the same capacity. [27] applied a dynamic discrete choice model to predict the timing in which ticket exchange or cancellation occurs in response to fare and trip schedule uncertainty. [28, 29] put the resource capacity allocation and fare rate of high-speed railway together for collaborative optimization. [30] allowed passengers to choose between the principle of time minimization and cost minimization, and established a balanced passenger flow assignment model based on personalized selection. [31] studied the differential pricing of intercity high-speed railway by analyzing the distribution characteristics of passenger demand and simulating the process of ticket purchase. [32] established a bilevel programming model under static condition. [33] proposed a non-concave nonlinear mixed integer optimization model of fare and seat allocation, which was solved by relaxation iteration. [34] proposed an improved machine learning gradient boosting decision tree to optimize the classification of price discrimination, and constructed a dynamic pricing framework based on Markov characteristics of passengers.

Table 1. Relevant literature research content statistics.

| References | Revenue management | Dynamic pricing strategy | Passenger choice behavior | Passenger flow assignment | Transport product classification |
|---|---|---|---|---|---|
| [4–6] | Y | Y | N | N | N |
| [7, 8] | Y | N | N | N | N |
| [9, 10, 12, 14, 19, 22, 23] | N | N | Y | N | N |
| [11, 13, 15–18, 20, 21, 24, 25] | N | N | Y | Y | N |
| [26–29, 32, 33] | Y | N | Y | N | N |
| [30, 31] | Y | N | Y | Y | N |
| [34] | Y | N | Y | N | Y |
| This paper | Y | Y | Y | Y | Y |

Whether research content is included: Consider (Y); Not considered (N).

The existing literatures have been comprehensively studied, including single-field and cross-field studies. The comparison of existing literatures is shown in Table 1.

From an academic research perspective, there is still room for further research of the price adjustment ideas and schemes at the operational level.

From the perspective of dynamic pricing, the ideal dynamic pricing model produced by the existing research can not fully adapt to the actual pricing work, and it is difficult to ensure the expected economic and social benefits. The positioning of high-speed railway transport products is rarely considered. Train classification can be combined to optimize the operability of the dynamic pricing strategy.

From the perspective of passenger flow assignment, some studies did not consider passenger flow assignment, and some studies assign passenger flow through relatively simple algorithm (such as k-short algorithm, all-or-nothing algorithm, successive averaging algorithm, etc.). The passenger flow assignment algorithm can be optimized to achieve accurate train-flow matching.

The innovations of this study are as follows:

1. Taking the game competition relationship and revenue management of differentiated train products into consideration, dynamic pricing is combined with passenger flow assignment. The train classification is combined linked with specific ticket price strategy, and different price adjustment ranges are set for different classes of trains.

2. The path size logit assignment based on capacity constraint-passenger flow increment accurate algorithm is nested into the large-scale neighborhood search algorithm. The exact matching of passenger flow and train flow is ensured while the heuristic iteration of ticket price is carried out. Real high-speed railway operational data are entered in case study, including passenger flow data and train timetable data.

## 3 Key notations definition

Considering the full manuscript involves many symbols and parameters, and their locations are scattered, the definitions of key symbols are listed centrally in Table 2 for readability and understanding.

**Table 2. Notations definition table.**

| Notation | Definition | Remark |
|---|---|---|
| *OD* | Passenger flow ODs set | $w \in OD$ |
| *Route* | Reasonable service routes set, containing direct routes(i.e. one train) and transfer routes(i.e. multiple trains) | $k, j \in$ *Route* |
| *Seat* | Train seats set | $s \in Seat$ |
| *Sta* | Station set | $i, j, l \in Sta$ |
| $C_{w,k,s}$ | The generalized cost of the $k$ option (i.e. route $k$) for seat $s$ of OD $w$ | Section 4 |
| $FC_{w,k,s}$ | The fixed cost of the $k$ option for seat $s$ of OD $w$ | |
| $RCC_{w,k}$ | The route correlation cost of the $k$ option of OD $w$ | |
| $v$ | The random cost of reasonable service routes | |
| $TC_{w,k}$ | The time cost of fixed cost | |
| $MC_{w,k}$ | The money cost of fixed cost | |
| $SC_{w,k,s}$ | The seat comfort cost of fixed cost | |
| $\alpha, \beta, \gamma$ | The parameters of fixed cost, $\alpha + \beta + \gamma = 1$ | |
| $\tau_{w,k}$ | The travel time in transit of the $k$ option of OD $w$ | |
| $n_{w,k}$ | The transfer times of the $k$ option of OD $w$ | |
| $\psi_{w,k}$ | The transfer penalty coefficient of the $k$ option of OD $w$ | |
| $m_{w,k}$ | The ticket price of the $k$ option of OD $w$ | |
| $\kappa_w$ | The passenger time value of OD $w$ | |
| $GDP$ | The regional economic development level | |
| $P$ | The regional population | |
| $LT$ | The regional per capita annual labor time, taking 2000 hours per year | |
| $LR$ | The regional average labor ratio, taking 0.7 | |
| $\eta$ | The passenger ticket price sensitivity coefficient | |
| $M$ | The maximum fatigue recovery time, $M = 15$, unit is hour | |
| $\tau$ | The dimensionless parameter, $\tau = 59$ | |
| $\delta$ | The strength coefficient of fatigue recovery time, $\delta = 0.28$, unit is h$^{-1}$ | |
| $\mu$ | A positive scale parameter, which can be 1 if there is a lack of data fitting | |
| $l_g$ | The length of route section $g$ | |
| $L_k$ | The total length of route $k$ | |
| $\lambda_{j,g}$ | Variable 0–1. If link $g$ belongs to route $j$, $\lambda_{j,g} = 1$ | |
| $U_{w,k,s}$ | The passenger travel utility of the $k$ option for seat $s$ of OD $w$ | |
| $C_{w,k,s}^{min}$ | The minimum generalized cost for seat s of OD w | |
| $C_{w,k,s}^{max}$ | The maximum generalized cost for seat $s$ of OD $w$ | |
| $f_{w,k,s}$ | The passenger flow assigned by OD $w$ to seat $s$ in route $k$ | Section 5 |
| $p_{w,k,s}$ | The ticket price of OD $w$ to seat $s$ in route $k$ | |
| $\varphi$ | The maximum decrease proportion of ticket price | |
| $\delta$ | The maximum increase proportion of ticket price | |
| $p_{w,k,s}^0$ | The base ticket price of OD $w$ to seat $s$ in route $k$ | |
| $n$ | The number of station intervals of OD $w$ in route $k$ | |
| $l$ | The OD corresponding to the adjacent station interval, $l \in [1,n]$, $l \in Z^+$ | |
| $p_{l,k,s}$ | The ticket price of OD $l$ to seat $s$ in route $k$ | |
| $\delta_{i,j}^{w,k}$ | Variable 0–1. If route $k$ of OD $w$ contains space-time arc $(i,j)$, $\delta_{i,j}^{w,k} = 1$. | |
| $Cap_{i,j}$ | The capacity of space-time arc $(i,j)$, $Cap_{i,j} \in N$ | |
| $Q_w$ | The total passenger flow of OD $w$ | |
| $h_{w,k,s}$ | The transfer times of OD w to seat s in route k | |
| $Prob_{w,k,s}$ | The selection ratio of the $k$ option for seat $s$ of OD $w$ | |
| $\theta$ | The choice probability parameter of path size logit model | |

# 4 Passenger flow assignment and passenger choice behavior

## 4.1 Railway passenger flow assignment overall process

Railway passenger flow assignment is the process of assign passenger flow to different trains according to certain principles and methods. It is an important link in the planning and design of passenger transport products.

Through passenger flow assignment, the train matching relationship under given conditions can be obtained, and then the passenger transport products can be evaluated and optimized, which can effectively improve the utilization level of railway resources and service level, and has important theoretical and practical significance.

The high-speed railway passenger flow assignment process is mainly composed of three parts, respectively route selection, model selection, passenger-train match, as shown in Fig 1.

## 4.2 Construction of train service network based on timetable

The whole travel process of railway passengers mainly consists of several stages: waiting on stations, getting on trains, moving in trains, transferring if needed, getting off trains and going out of stations. The 2nd to 5th stages are selected to construct the service network which contained a sequence of service nodes and arcs.

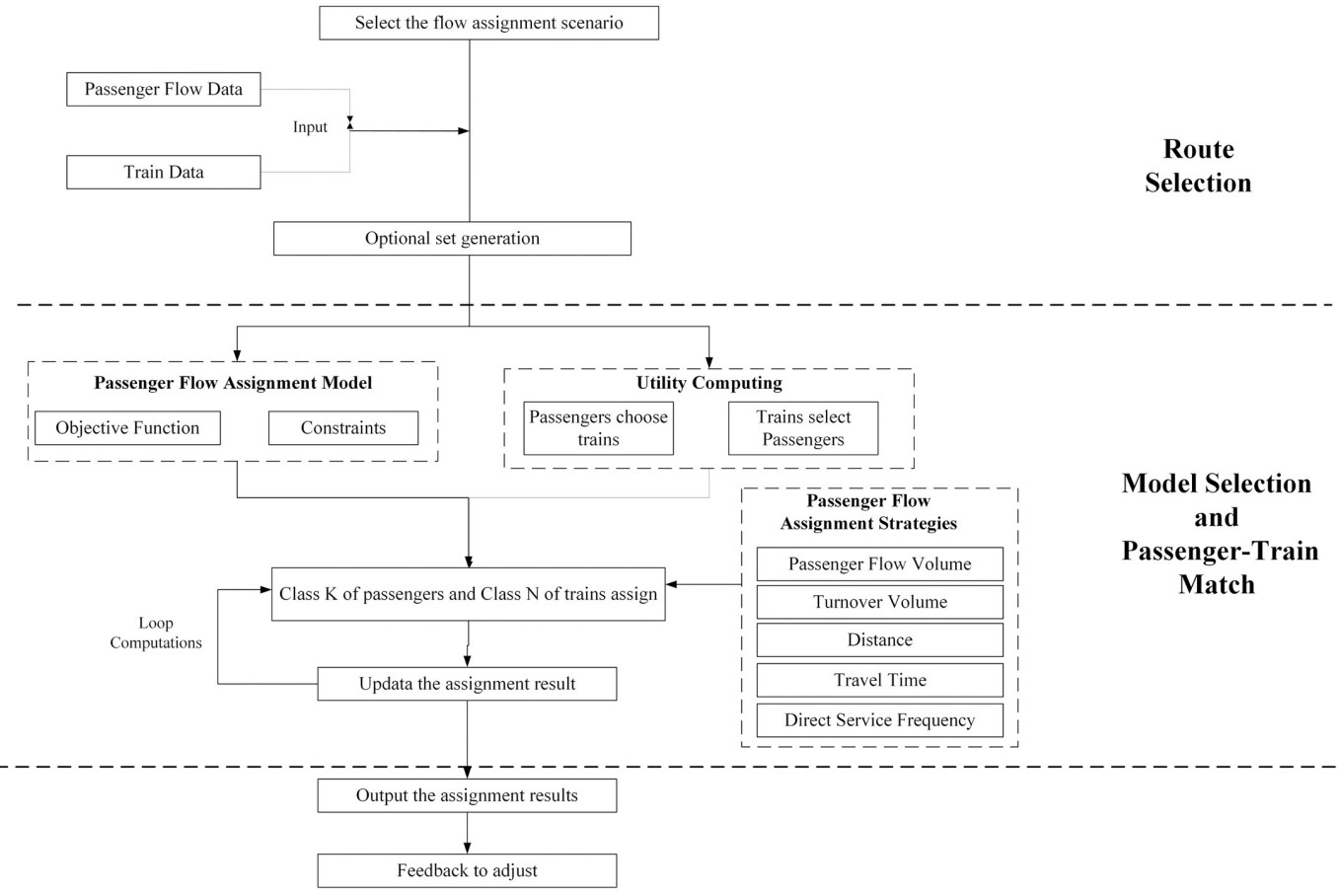

**Fig 1. High-speed railway passenger flow assignment process.**

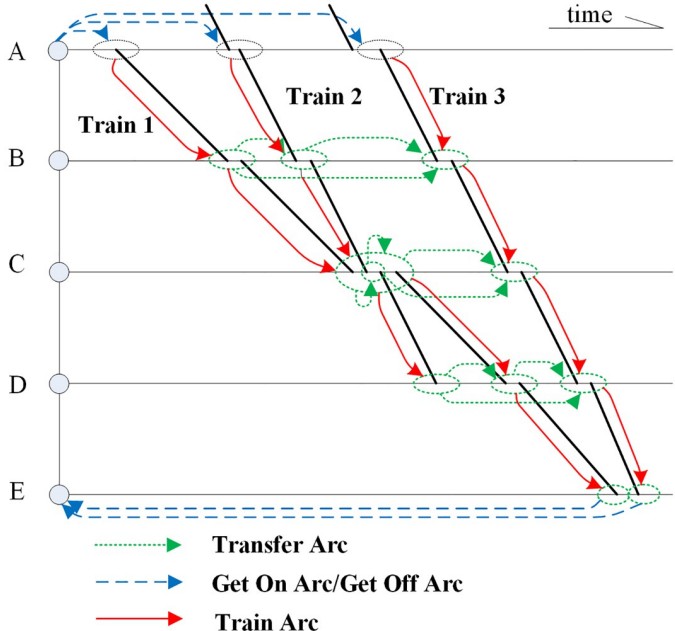

**Fig 2. Sample of train service network.**

Service nodes comprise of one or several stations in the same city or special area where trains depart, pass or arrive and passengers can transfer from one train to another. The arcs are used to describe all the actions of passengers including getting on/off the trains, moving by the trains and transfer between the trains. A small service network comprise of 3 trains and 5 nodes (***A-E***) are shown in Fig 2 where the horizontal direction indicates time and the vertical direction indicates space.

The service network is represented by ***G*** = {***Node, Arc***}, where ***Node*** is the set of nodes and ***Arc*** = {$e_{ft,st,n}$} is the set of all kinds of arcs. The symbol $e_{ft,st,n}$ represents 4 types of arcs:

1. Getting on arc by which passengers get on train ***ft*** from node ***n*** when ***st*** = 0;

2. Getting off arc by which passengers get off train ***st*** from node ***n*** when ***ft*** = 0;

3. Moving arc by which passengers pass through the ***st***$^{th}$ section of train ***ft*** when ***n*** = 0;

4. Transfer arc by which passengers transfer from train ***ft*** to ***st*** in node ***n***.

Then several sample routes in Fig 2 from node ***A*** to node ***E*** can be presented as following.

1. Two Direct routes: {$e_{2,0,a}$, $e_{2,1,0}$, $e_{2,2,0}$, $e_{2,3,0}$, $e_{2,4,0}$, $e_{0,2,e}$}, {$e_{3,0,a}$, $e_{3,1,0}$, $e_{3,2,0}$, $e_{3,3,0}$, $e_{3,4,0}$, $e_{0,3,e}$};

2. Eight one-time transfer route: {$e_{2,0,a}$, $e_{2,1,0}$, $e_{2,3,b}$, $e_{3,2,0}$, $e_{3,3,0}$, $e_{3,4,0}$, $e_{0,3,e}$}, {$e_{2,0,a}$, $e_{2,1,0}$, $e_{2,2,0}$, $e_{2,3,c}$, $e_{3,3,0}$, $e_{3,4,0}$, $e_{0,3,e}$}, {$e_{2,0,a}$, $e_{2,1,0}$, $e_{2,2,0}$, $e_{2,3,0}$, $e_{2,3,d}$, $e_{3,4,0}$, $e_{0,3,e}$}, etc;

In summary, there are 14 travel routes for the passengers from node A to E including another 4 two-time transfer routes. The quantity of service routes in such a simple service network is evidently a little much, and it will go up very sharply with the increase of the number of trains in the network.

In order to reduce the scale of train service network search, trains are converted to nodes and the main part of abstracted service network is comprised of trains nodes and transfer arcs.

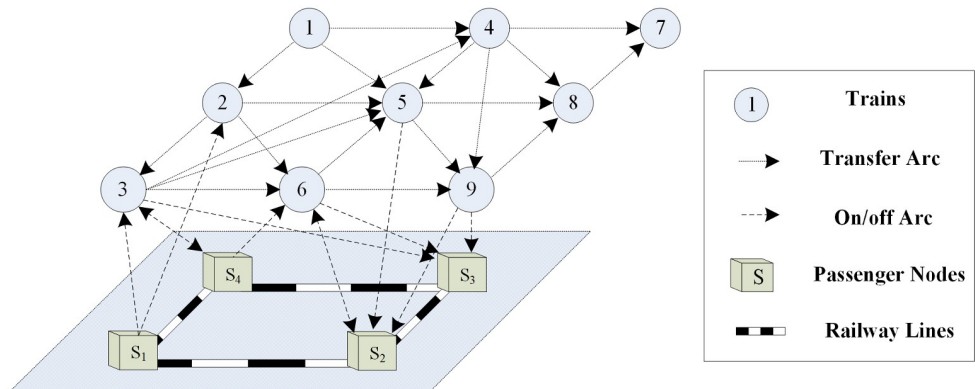

**Fig 3. Abstracted train service network.**

In addition, passengers enter or exit the network by passenger nodes and getting on/off arcs, as shown in Fig 3. Figs 2 and 3 are intended to illustrate the different forms of train service network, and there is no correspondence between them.

*TransferArc* = {$e_{ft,st}$} is the set of transfer arcs, where *ft* represents the first train, *st* represents the second train. *OnArc* = {$e_{n,ontrain}$} is the set of getting on arcs, where *n* represents the getting on passenger node and *ontrain* represents the getting on train. *OffArc* = {$e_{n,offtrain}$} is the set of getting off arcs, where *n* represents the getting off passenger node and *offtrain* represents the getting off train. The time attributes of passenger getting on/off and transfer are included in the sub-attributes of trains and are not represented in the abstracted train service network.

## 4.3 Generalized cost formula of reasonable service routes

In the space-time service network, there are a large number of available routes for same OD at the same time, but in the actual travel process, some service routes will not be chosen. Therefore, the reasonable service route is defined as the OD service route that meets the passenger travel demand and conforms to the relevant business rules and common sense.

The passenger choice of service route affected by several factors including total travel time, total ticket price, convenience of the journey, departure and arrival time, etc. Passengers would generally choose the routes with the minimum cost in their minds.

[9] noted that the path over-lapping problem violates the IIA irrelevant independence hypothesis of *MNL* model and further affects the passenger route selection behavior. Therefore, *PSL* model is used to replace *MNL* model, in which the route generalized cost is divided into fixed cost, route correlation cost and random cost.

From the perspective of passengers, the generalized cost formula of reasonable service routes is defined as follows.

$$C_{w,k,s} = FC_{w,k,s} + RCC_{w,k} + v \tag{1}$$

$$FC_{w,k,s} = \alpha \cdot TC_{w,k} + \beta \cdot MC_{w,k} + \gamma \cdot SC_{w,k,s} \tag{2}$$

$$TC_{w,k} = \tau_{w,k} + \psi_{w,k} \cdot n_{w,k} \tag{3}$$

**Table 3. Calculation of passenger time value.**

| Region | GDP(*trillion yuan*) | P(*million*) | LT | LR(*h*) | $\kappa_w$(*yuan/h*) |
|---|---|---|---|---|---|
| Shandong province | 7.05 | 10070 | 0.66 | 2190 | 48.44 |
| China | 98.6515 | 141000 | 0.64 | 2190 | 49.92 |

Data source: National Bureau of Statistics for 2019, Shandong Statistical Yearbook for 2019, Statistical Bulletin of the National Economic and Social Development of the People's Republic of China for 2019, Statistical Bulletin of the National Economic and Social Development of Shandong Province for 2019.

$$MC_{w,k} = m_{w,k}/\kappa_w \tag{4}$$

$$\kappa_w = \frac{GDP}{P \cdot LT \cdot LR} \tag{5}$$

$$SC_{w,k,s} = \eta \cdot \frac{M}{1 + \tau e^{-\delta \cdot TC_{w,k}}} \tag{6}$$

In terms of calculating values, each subitem of the fixed cost $FC_{w,k,s}$ are uniformly transformed into time to calculate.

Time cost $TC_{w,k}$ is related to the travel time in transit and convenience of the journey which can be expressed as transfer time, as shown in Eq (3). The transfer penalty coefficient $\psi_{w,k}$ takes 0.5 hours per time.

Money cost $MC_{w,k}$ is related to the total ticket price that can be converted into time cost according to the passenger time value, as shown in Eqs (4) and (5).

Since the *Shandong circular high-speed railway* not only includes the passenger flow in Shandong province, in order to ensure the availability of data, the passenger time value of China and Shandong province were calculated respectively. Table 3 is the calculation of passenger time value.

Table 4 is the Passenger flow of Jinan Railway Bureau in 2019. The local passengers and export passengers correspond to the passenger time value of Shandong province. The import passengers and carrying passengers correspond to the passenger time value of China.

According to the proportion of different types of passenger flow, the passenger time value was 49.18 yuan/h, which was rounded up to 50 yuan/h.

Seat comfort cost $SC_{w,k,s}$ is related to the fatigue recovery time, as shown in Eq (6). For high-speed railway trains [35], defined the values of $M$, $\tau$ and $\delta$ as 15, 59 and 0.28 respectively. Based on the actual ticket price approximate ratio of the same OD and same train, the passenger fare sensitivity coefficient $\eta$ for business seat, first class seat and second class seat are 3, 1.5, 1, respectively.

**Table 4. Passenger flow of Jinan Railway Bureau in 2019.**

| Passenger flow type | Number(*million*) |
|---|---|
| Local | 9192 |
| Export | 6011 |
| Import | 6050 |
| Carrying | 9241 |

Data source: Compilation of National Railway Statistical Data in 2019.

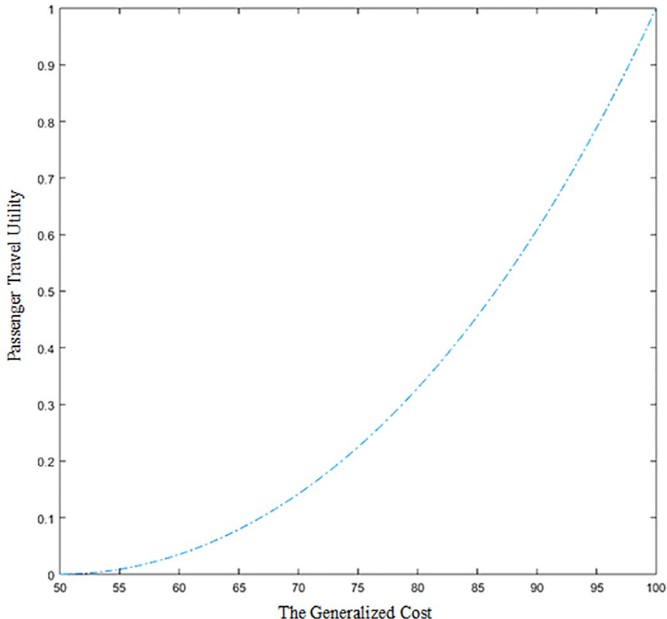

**Fig 4. Chart of changes in passenger travel utility.**

[9] deduced three formulas of route choice correlation and pointed out that the original formulation should be preferred, which has a theoretical foundation.

$$RCC_{w,k} = \mu \ln \sum \frac{l_g}{L_k} \frac{1}{\sum\limits_{j \in R} \lambda_{j,g}} \tag{7}$$

On the basis of Eq (1), the passenger travel utility formula is defined as follows. The purpose of *arcosh* (2) is to guarantee $U_{w,k,s} \in [0,1]$.

$$U_{w,k,s} = \cosh\left(\frac{C_{w,k,s} - C_{w,k,s}^{min}}{C_{w,k,s}^{max} - C_{w,k,s}^{min}} \bullet arcosh(2)\right) - 1 \ C_{w,k,s} \in [C_{w,k,s}^{min}, C_{w,k,s}^{max}] \tag{8}$$

When $C_{w,k}^{max} = 100$, $C_{w,k}^{min} = 50$, the relationship between passenger travel utility and generalized cost of reasonable service routes is shown in Fig 4. As can be seen from the slope of the curve in Fig 4, the lower the generalized cost value, the more difficult it is to change the passenger travel utility.

## 5 Multi-objective dynamic pricing model and algorithm

Railway 12306 website (including mobile client) is the only official online ticketing platform of China Railway. When passengers buy tickets, they need to correspond with their ID cards one by one. Even group travel tickets are displayed in the 12306 system as separate ticket purchases. Additionally, Chinese high-speed railway only allows a very limited number of trains to carry a few extra passengers during busy holidays.

Based on the above national conditions, the following assumptions are made:

1. The model does not consider overbooking.

2. The travel needs of passengers are independent and do not affect each other.

## 5.1 Objective function

The first optimization objective is to maximize the corporate revenue. Let $M$ denote the total revenue of high-speed railway enterprises and $Z_1$ denote the revenue-related optimization objective.

$$M = \sum_w \sum_k \sum_s f_{w,k,s} p_{w,k,s} \tag{9}$$

$$Z_1 = \max M \tag{10}$$

Let $N$ denote the passenger travel choice benefit and $Z_2$ denote the passenger-related optimization objective.

$$N = \sum_w \sum_k \sum_s f_{w,k,s} C_{w,k,s} \tag{11}$$

$$Z_2 = \min N \tag{12}$$

The methods for solving multi-objective programming problems include transformation constraint method, evaluation function method and hierarchical sequence method. In the absence of historical data, it is difficult for either $Z_1$ or $Z_2$ to set the initial constraint range, so one of $Z_1$ or $Z_2$ cannot be converted into a constraint for solving. Considering that total revenue represented by $Z_1$ and passenger travel choice benefit represented by $Z_2$ are difficult to be treated as the same evaluation function.

To sum up, the hierarchical sequence method is selected. Firstly, the ticket price in $Z_1$ is dynamic by heuristic algorithm (*Large Neighborhood Search*). Secondly, the precise algorithm (*Path Size Logit Passenger Flow Assignment*) is used to simulate passenger travel choice behavior and solve $Z_2$ passenger flow assignment. Finally, the $Z_2$ result is substituted into $Z_1$ to solve the total revenue.

## 5.2 Decision variable and constraints

As the decision variable of dynamic pricing problem, the ticket price $p$ is related to the line price rate per unit distance $p_0$ and the distance of current OD. $p$ changes with the change of $p_0$, and then changes the reasonable service route utilities $c$ and passenger flow assignment results $f$. It should be noted that since the reasonable service routes contains direct routes and transfer routes, the subscript of $p$ uses route (i.e. subscript $k$) instead of train.

$Z_1$ considers multiple constraints which are related to ticket price, including dynamic pricing constraint, price not inverted constraint, as shown in Eqs (13) and (14).

1. **Dynamic pricing constraint.** In terms of ticket price related constraints, the dynamic range of price is restricted. Too high a price is not conducive passenger travel, and too low a price is not conducive corporate revenue.

$$p_{w,k,s}^0 (1 + \varphi) \leq p_{w,k,s} \leq p_{w,k,s}^0 (1 + \delta) \tag{13}$$

2. **Price not inverted constraint.** Ticket prices should not be reversed, which means the price of a short-distance train ticket is lower than the price of a long-distance train ticket. If the total distance is equal, the combined value of two short-distance train ticket is not less than

that of one long-distance train ticket.

$$\sum_l p_{l,k,s} \geq p_{w,k,s}, p_{l,k,s} \leq p_{w,k,s}, l \in [1,n] \tag{14}$$

$Z_2$ considers multiple constraints which are related to passenger flow, including passenger choice route capacity constraint, route passenger flow non-negative constraint, OD passenger flow identically equal constraint, transfer times constraint, as shown in Eqs (15)–(18).

3. **Passenger choice route capacity constraint.** Each reasonable service route has a maximum passenger flow capacity, which is limited by the capacity of the trains serving the space-time service route.

$$\sum_w \sum_k \sum_s f_{w,k,s} \delta_{i,j}^{w,k} \leq Cap_{i,j} \tag{15}$$

4. **Route passenger flow non-negative constraint.** In terms of passenger flow related constraints, the passenger flow of any seat level on each route must remain non-negative be negative at any time in any OD.

$$f_{w,k,s} \geq 0 \tag{16}$$

5. **OD passenger flow identically equal constraint.** According to common sense, the sum of passenger flow on all routes at any time is a fixed value.

$$\sum_k \sum_s f_{w,k,s} = Q_w \tag{17}$$

6. **Transfer times constraint.** When the capacity of direct service routes is insufficient, passengers will tend to choose transfer service routes. However, too many transfer times will against the travel experience of passengers. According to the investigation of the maximum passenger acceptable transfer times in relevant areas and the passenger flow analysis of *Shandong circular high-speed railway*, the maximum acceptable transfer times is set to be no more than 1.

$$h_{w,k,s} \leq 1 \tag{18}$$

## 5.3 Large neighborhood search-path size logit passenger flow assignment algorithm

In this paper, the problem of dynamic pricing and passenger flow assignment of high-speed railway is transformed into a combination optimization problem of maximizing passenger travel choice benefit under the condition of maximizing corporate revenue.

The path size logit passenger flow assignment accurate algorithm is embedded into the structure of large neighborhood search heuristic algorithm. The overall algorithm structure and calculation process are shown in Fig 5. The pseudo-code of large neighborhood search (*LNS*) algorithm is shown in Table 5.

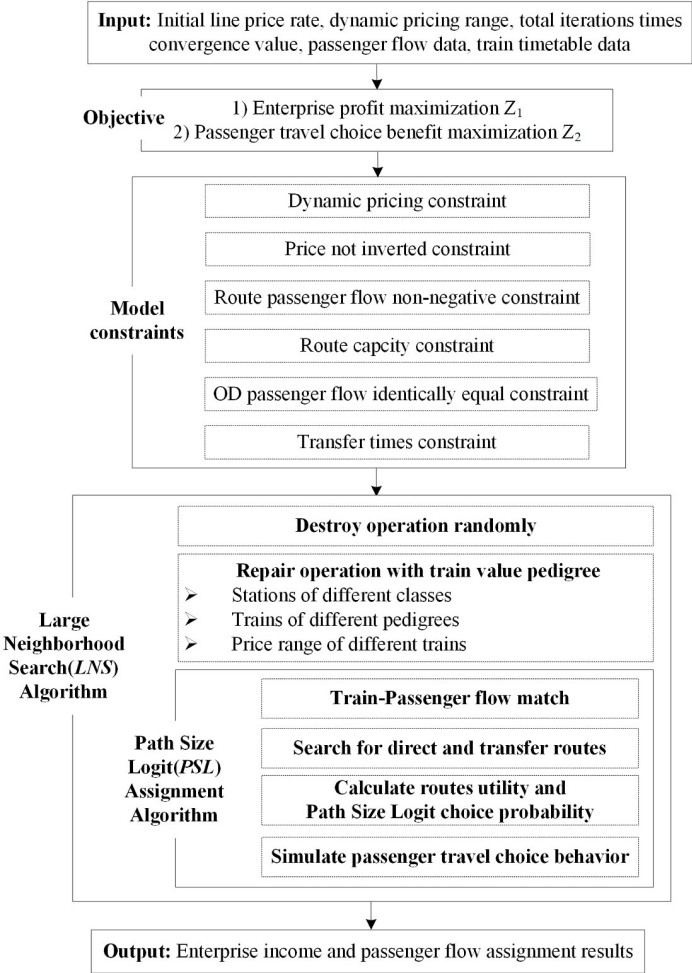

**Fig 5. The overall algorithm structure.**

Step 1: Input initial data, including line price rate $R_0$, dynamic pricing range [*LowerRate*, *UpperRate*], total iterations times $T$, convergence value $\Delta c$, OD passenger flow data, train timetable data.

Step 2: Establish the classification of trains based on multiple factors (such as number or proportion of different level stations, etc.). Single or multiple factors are considered, depending on the availability of relevant data.

Step 3: Establish the subdivision ticket price adjustment strategy based on train class.

- Step 3.1: Set different price gaps. The positions of different gaps are different. The practical significance of the narrow gap is greater, and the research significance of the wide gap is greater.

- Step 3.2: The ticket price adjust ranges of different class trains under different gaps are determined. The higher the train class, the higher the ticket price rate.

Step 4: Carry out the large neighborhood search (*LNS*) algorithm.

**Table 5. Pseudo-code of large neighborhood search (*LNS*) algorithm.**

| Large Neighborhood Search (*LNS*) Algorithm |
| --- |
| 1: input: dynamic pricing range [*LowerRate*, *UpperRate*], total iterations times $T$, current iterations times $t$, train level judging standard, initial line price rate $R_0$, convergence value $\Delta c$ |
| 2: **While** $t \leq T$ |
| 3:    **if** $t = 0$ **then** |
| 4:        current global optimal solution $x^b = x_0$ (initial feasible solution and $x_0 = 0$); |
| 5:    **else then** |
| 6:        current global optimal solution $x^b = x_1$ (output of last iteration); |
| 7:    **end if** |
| 8:    **Destroy:** select $P$ percent of trains randomly as $F$, P∈(0,1); |
| 9:        **if** a train is selected repeatedly **then** |
| 10:            **continue;** |
| 11:        **end if** |
| 12:    **Repair:** adjust price rate $R$ randomly of $F$ based on train levels; |
| 13:        **for** train $f \in F$ |
| 14:            Based on station level, judgment matrix and price adjustment strategy, judge train classes |
| 15:                (Class I, Class II, Class III) and corresponding price range which is a subset of [*LowerRate*, *UpperRate*]; |
| 16:        **end for** |
| 17:    **Path Size Logit Assignment Algorithm** (*PSL*, as shown in Fig 6) |
| 18:    **Calculate current iteration total revenue $x$** |
| 19:    **Evaluate** |
| 20:        **if** $\mid x - x^b \mid \leq \Delta c$ **then** |
| 21:            **break while** and **return** $x^b$; |
| 22:        **else then** |
| 23:            **if** E($x$)>E($x^b$) **then** |
| 24:                update $x^b = x$; |
| 25:            **end if** |
| 26:        **end if** |
| 27:    **return** $x^b$; |
| 28: **end While** |

- Step 4.1: Select **P** percent of the trains randomly for **Destroy** and **Repair** operations. The value of **P** can be deterministic or random.

- Step 4.2: **Destroy** means the selected trains (**P** percent quantity and reserved integer) are removed from the train alternative set. The ticket price rate of the selected trains will be adjusted randomly within the range by the above strategy. The ticket price rate of other trains in the alternative set remains unchanged.

- Step 4.3: **Repair** means the ticket price rate of the selected trains has been adjusted. And then they are inserted into the train alternative set.

Step 5: Carry out the path size logit (*PSL*) assignment algorithm based on capacity constraint-passenger flow increment (*CCPFI*).

- Step 5.1: Search for direct routes and transfer routes for different ODs based on train time-table data.

- Step 5.2: For different routes of different ODs, calculate route utility and path size logit choice probability based on train price rate (i.e. the route selection ratio).

- Step 5.3: Considering OD classes, OD passenger flow demand and train capacity, simulate passenger travel choice behavior. If the train capacity is sufficient, the number of passengers on board is the product of route selection ratio and OD passenger flow demand. Otherwise, the number of passengers on board is the train surplus capacity.

- Step 5.4: Output enterprise income $x$ and passenger flow assignment results.

Step 6: Evaluate the output result against the current global optimal solution $x^b$. Determine whether the iteration times $T$ is reached. If true, go Step 9. Otherwise, go Step 7.

Step 7: If $|x—x^b|\leq\Delta c$, go Step 9. Otherwise, go Step 8.

Step 8: If $x\geq x^b$, update the current global optimal solution $x^b = x$. Then, go Step 4.

Step 9: End the loop and output $max\{x, x^b\}$.

According to the random utility theory of passenger choice behavior, path size logit assignment algorithm based on capacity constraint-passenger flow increments calculated using Eq (19) where $U_{w,k,s}$, $U_{w,m,s}$ are given in Eq (8). The specific process is shown in Fig 6.

$$Prob_{w,k,s} = \frac{\exp(-\theta U_{w,k,s})}{\sum\limits_{m} \exp(-\theta U_{w,m,s})} \tag{19}$$

## 6 Case study

### 6.1 Basic information of *Shandong circular high-speed railway*

The *Shandong circular high-speed railway* consists of 5 railway lines, namely the *Jiqing HSR* (East-West direction), *Jiaoji HSR* (East-West direction), *Jinghu HSR* (North-South direction), *Rilan HSR* (East-West direction), and *Qingyan HSR* (Northeast-Southwest direction). The schematic diagram of *Shandong circular high-speed railway* based on station levels is shown in Fig 7.

Based on the number of trains served, the number of passengers served and the city administrative level, the stations are divided into three levels, including large station, medium station and small station. For *Shandong circular high-speed railway*, there are 33 stations, including 7 large stations, 8 medium stations and 18 small stations, as shown in Fig 7.

It is worth noting that Jinan and Qingdao are the key traffic nodes of *Shandong circular high-speed railway*, both of which have multiple large stations. Although *Qingdao West* belongs to Qingdao administratively, the passenger flow is relatively small, so it is classified as a medium station.

*Linyi North* is located in the center of southern part of Shandong province, so it is classified as a large station. *Rizhao West* is a transit node connecting Shandong province and Jiangsu province. Although the passenger flow is lower than Jinan node and Qingdao node, it is classified as a large station considering its special geographical location. In addition, due to the close distance between *Jiaozhou North* and *Qingdao Airport*, they are merged into a small station.

Based on the historical ticket data, the average basic ticket price rate of *Shangdong circular high-speed railway* is set at 0.4 *yuan/km*. In fact, the ticket price rate of different sections of the same line may be different, and high-speed railway has the characteristics of decreasing rate with increasing distance. However, the above points do not affect case design and validation of the model and algorithms.

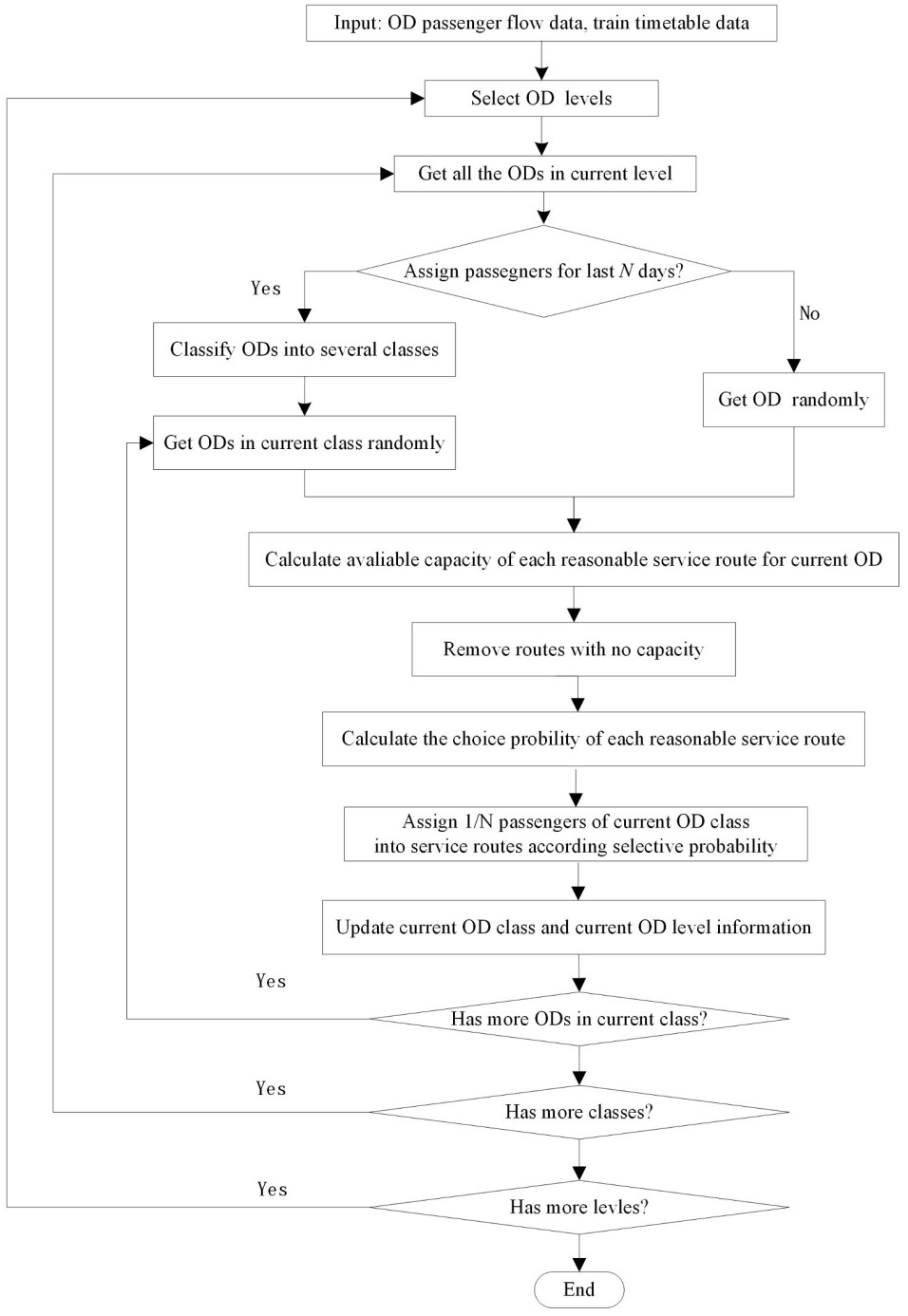

**Fig 6. Path size logit assignment based on capacity constraint-passenger flow increment algorithm.**

The passenger flow data and train timetable data on October 25 in 2019 were provided by the Jinan Railway Bureau.

Due to the availability of data, only second-class seats are considered in case. It should be noted that the research approach for different classes of seating is the same, with the emphasis on distinguishing the heterogeneity of the target passengers. The acceptable range of dynamic

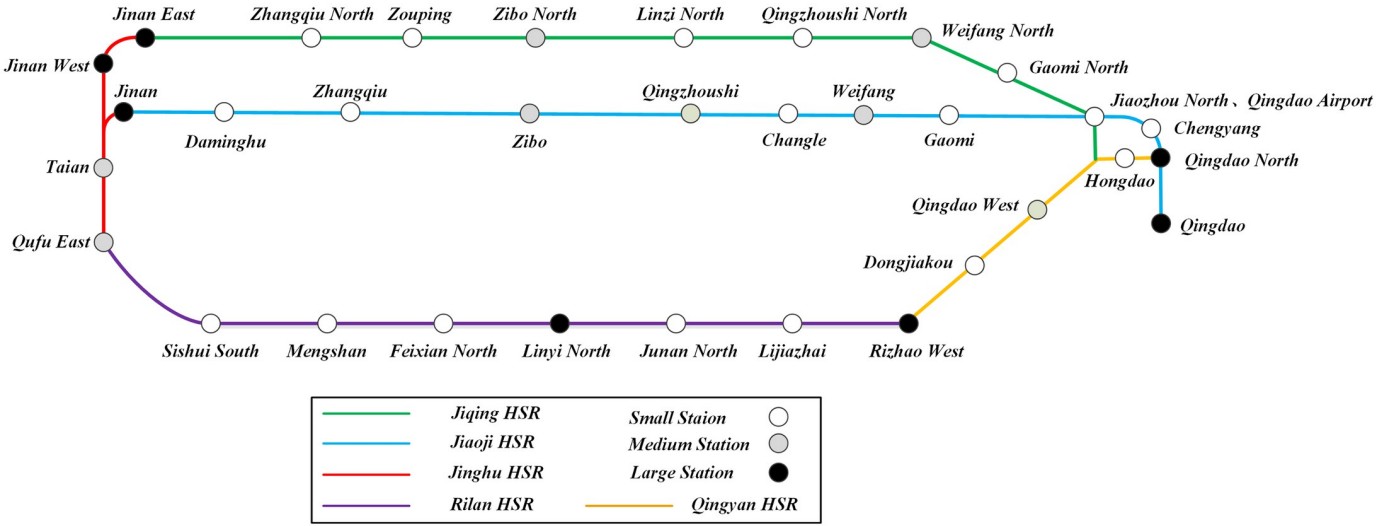

**Fig 7. The schematic diagram of *Shandong circular high-speed railway*.**

pricing varies among different passengers. Therefore, the second-class seat scenario case can still verify the applicability of the model and algorithms which can still be applied to the scenarios considering different seats.

## 6.2 Train classification and price adjustment strategy

The process of railway transportation organization is very complex, involving railway network planning, line plan, train timetable, EMU plan, crew plan, station organization, marketing organization, etc. Each link is nested and affects each other.

In order to maintain the stability of the railway system and improve the social and economic benefits of high-speed railway, the classification process of differentiated train products should consider both scientific and practical, which means that the classification criteria should be scientifically grounded but not overly complex.

[36] described a route synthesis layering approach that defined stations and trains according to two classes, which contained the idea of classification. On this basis, the train classification operation is as follows: Firstly, the station level is divided based on the historical average passenger flow and the city administrative level. Secondly, differentiated train products are divided into Class I, Class II and Class III based on the number of stations and station level. Finally, the train classification is linked to ticket price. Different classes of trains have different dynamic pricing range.

Generally speaking, there are two optional criteria for the establishment of train classification. First, it is divided by the station levels of train service, such as the large station train only serves the large station, the medium station train only serves the large station and the medium station, and the small station train must serve the small station. Second, it is divided by the proportion of stations at all levels of train service, such as the proportion of $x$ stations in all stations $p^x$, where $x \in \{large, medium, small\}$.

Due to the large number of small stations along the *Shandong circular high-speed railway*, all trains serve the above three levels of station. It is impossible to construct the train classification according to the first criteria.

**Table 6. Train classification decision table.**

| Train Class | Decision Matrix [$p^{large}$-$p_1$, $p^{medium}$-$p_2$, $p^{small}$-$p_3$] | | |
|---|---|---|---|
| Class I | 1 | 1 | 0 |
| | 1 | 0 | 0 |
| Class II | 1 | 0 | 1 |
| | 0 | 1 | 0 |
| Class III | 0 | 1 | 1 |
| | 0 | 0 | 1 |

As can be seen from Fig 7, the proportion of large stations $p_1 \approx 21\%$, that of medium stations $p_2 \approx 24\%$, and that of small stations $p_3 \approx 55\%$. Based on the above ratio and $p^x$, the matrix [$p^{large}$-$p_1$, $p^{medium}$-$p_2$, $p^{small}$-$p_3$] is used as the train classification standard, where each item is a 0–1 variable. If the positive value is 1, otherwise 0. Since $p^{large}$+ $p^{medium}$ +$p^{small}$ = 100%, there is no case of [1,1,1] or [0,0,0]. The train classification determination table is shown in Table 6. Based on the decision criteria in Table 6, the different classes of trains is shown in Table 7.

Different price adjustment ranges are set based on different train class. When the train class is same and the decision matrix is different, the price adjustment ranges should be slightly adjusted. Taking the Class I as an example, due to the higher proportion of medium stations, the lower limit of the price adjustment range with the judgment matrix [1,1,0] is supposed to be higher than the lower limit of the price adjustment range with the judgment matrix [1,0,0], and the increase proportion ought to be related to the price adjustment gap.

The national conditions determine that high-speed railway enterprises must assume political tasks and social missions while pursuing the maximization of profits with the goal of marketization. The fixed ticket price mechanism formed under the government guidance for a long time involves many factors in railway transportation management, resulting in the current dynamic pricing mechanism is relatively strict control. The actual maximum price fluctuation gap in the eastern developed region is about 20%.

Based on this, we set price gap to a maximum of 40% and a minimum of 20%, with the same upper and lower bounds. The overall range based on train classification is half of the price gap. According to the station decision matrix, the subdivide range is half or all of the overall range. The specific ticket price adjustment strategies of *Shandong circular high-speed railway* are shown in Table 8.

**Table 7. Train classification table of *Shandong circular high-speed railway*.**

| Train | Large Stations | Medium Stations | Small Stations | All Stations | Decision Matrix | Train Class |
|---|---|---|---|---|---|---|
| G5525 | 4 | 5 | 5 | 14 | [1,1,0] | Class I |
| G5526 | 4 | 5 | 5 | 14 | [1,1,0] | Class I |
| G5535 | 3 | 2 | 6 | 11 | [1,0,1] | Class II |
| G5536 | 2 | 3 | 6 | 11 | [0,1,1] | Class III |
| G5555 | 2 | 4 | 4 | 10 | [0,1,0] | Class II |
| G5557 | 2 | 2 | 6 | 10 | [0,0,1] | Class III |
| G5565 | 5 | 4 | 3 | 12 | [1,1,0] | Class I |
| G5567 | 5 | 5 | 5 | 15 | [1,1,0] | Class I |
| G6955 | 2 | 4 | 8 | 14 | [0,1,1] | Class III |
| G6956 | 2 | 3 | 6 | 11 | [0,1,1] | Class III |
| G6975 | 2 | 4 | 6 | 12 | [0,1,0] | Class II |
| G6976 | 3 | 2 | 6 | 11 | [1,0,1] | Class II |

**Table 8. Specific ticket price adjustment strategy of *shandong circular high-speed railway*.**

| Price Gap | Train Class | Overall Range | Decision Matrix | Subdivide Range |
|---|---|---|---|---|
| 40% | Class I | 0 ~ 20% | [1,1,0] | 10% ~ 20% |
| | | | [1,0,0] | 0 ~ 20% |
| | Class II | -10% ~ 10% | [1,0,1] | 0 ~ 10% |
| | | | [0,1,0] | -10% ~10% |
| | Class III | -20% ~ 0 | [0,1,1] | -10% ~ 0 |
| | | | [0,0,1] | -20% ~ 0 |
| 30% | Class III | 0 ~ 15% | [1,1,0] | 7.5% ~ 15% |
| | | | [1,0,0] | 0 ~ 15% |
| | Class II | -7.5% ~ 7.5% | [1,0,1] | 0 ~ 7.5% |
| | | | [0,1,0] | -7.5% ~ 7.5% |
| | Class III | -15% ~ 0 | [0,1,1] | -7.5% ~ 0 |
| | | | [0,0,1] | -15% ~ 0 |
| 20% | Class I | 0 ~ 10% | [1,1,0] | 5% ~ 10% |
| | | | [1,0,0] | 0 ~ 10% |
| | Class II | -5% ~ 5% | [1,0,1] | 0 ~ 5% |
| | | | [0,1,0] | -5% ~ 5% |
| | Class III | -10% ~ 0 | [0,1,1] | -5% ~ 0 |
| | | | [0,0,1] | -10% ~ 0 |

Considering that there are differences in the output results of the heuristic algorithm each time, the large neighborhood search (*LNS*)-path size logit (*PSL*) assignment algorithm is set to cycle 100 times to 1000 times, and the average is taken as the result. In addition, the maximum number of iterations in a single loop to 20, and the convergence threshold $\Delta c = 1‰$.

## 6.3 Total revenue analysis

In order to verify the effectiveness of the model and algorithm under different price adjustment ranges (*Series1*[-10%,10%], *Series2*[-15%,15%], *Series3*[-20%,20%]), the **Destroy** and **Repair** operations ratio in the large neighborhood search is set to 50%, and the income curves under different price adjustment ranges are drawn respectively, as shown in Fig 8.

In terms of the revenue curve, with the increase of the ticket price adjustment range, the absolute income and the fluctuating range increases. The reason is that the upper and lower bounds of the ticket price floating range are the same, and they are gradually increasing.

In terms of average income, compared with *original price*, the average income increase rate of *Series1*, *Series2* and *Series3* is 3.63%, 5.9% and 7.92% respectively.

## 6.4 Ticket price comparison analysis

The basis of enterprise ticketing and passenger ticket purchase is the ticket price with ODs as the unit. Select "*Linyi North-Qingdao North*", "*Qingdao West-Qufu East*", "*Dongjiakou-Mengshan*" as the key ODs of different levels.

When the price adjustment range is set to [-20%,20%], the **Destroy** and **Repair** operations ratio in the large neighborhood search is set to 40%, remove unreasonable detour trains and calculate the price fluctuation range of different trains under various ODs. The average iterated ticket prices for key ODs of different classes are shown in Table 9. The box plots are drawn, as shown in Fig 9.

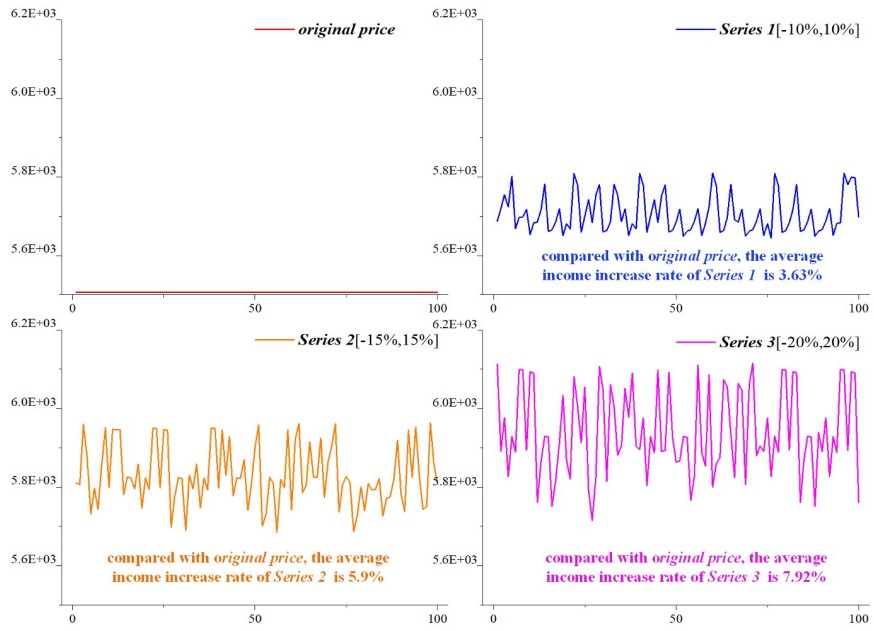

**Fig 8. Revenue comparison under different price adjustment ranges.**

In the box plots, the data above the box indicates that the size is sorted in the first 25%, the data below the box indicates that the size is sorted in the last 25%, the data box indicates that the size is sorted between 25%-75%. The upper boundary of the box represents the upper quartile and the lower boundary represents the lower quartile. The whiskers are lines extending from the box and cover 99.3% of the data volume. The upper and lower boundaries of the whiskers represent the maximum and minimum values of the data, respectively. MEAN value is the literal meaning. Outliers represent data which exceeds the whiskers.

The ticket price fluctuation of different trains(Class I *G5526* and Class III *G5536*) under large OD (*Linyi North-Qingdao North*) is calculated, as shown on the left side of Fig 9. In terms of ticket price dynamic range, the box range of *G5536* is obviously narrower than that of *G5526*, indicating that the ticket price of *G5536* is relatively stable and according with the basic situation that the ticket price fluctuation range of Class I is larger than that of Class III. In terms of MEAN value, the average ticket price of *G5536* is lower than that of *G5526*. In terms

**Table 9. Average iterated ticket prices for key ODs of different classes.**

| OD Level | DepSta | ArrSta | TrainNo | Ticket Price(*yuan*) |
|---|---|---|---|---|
| Large OD | *Linyi North* | *Qingdao North* | *G5526* | 103.3 |
| | | | *G5536* | 99.8 |
| | | | *G5567* | 110.9 |
| Medium OD | *Qingdao West* | *Qufu East* | *G5555* | 125.1 |
| | | | *G5565* | 130.6 |
| | | | *G6955* | 122.5 |
| | | | *G6975* | 122.8 |
| Small OD | *Dongjiakou* | *Mengshan* | *G5555* | 86.5 |
| | | | *G6955* | 84.2 |
| | | | *G6975* | 84.9 |

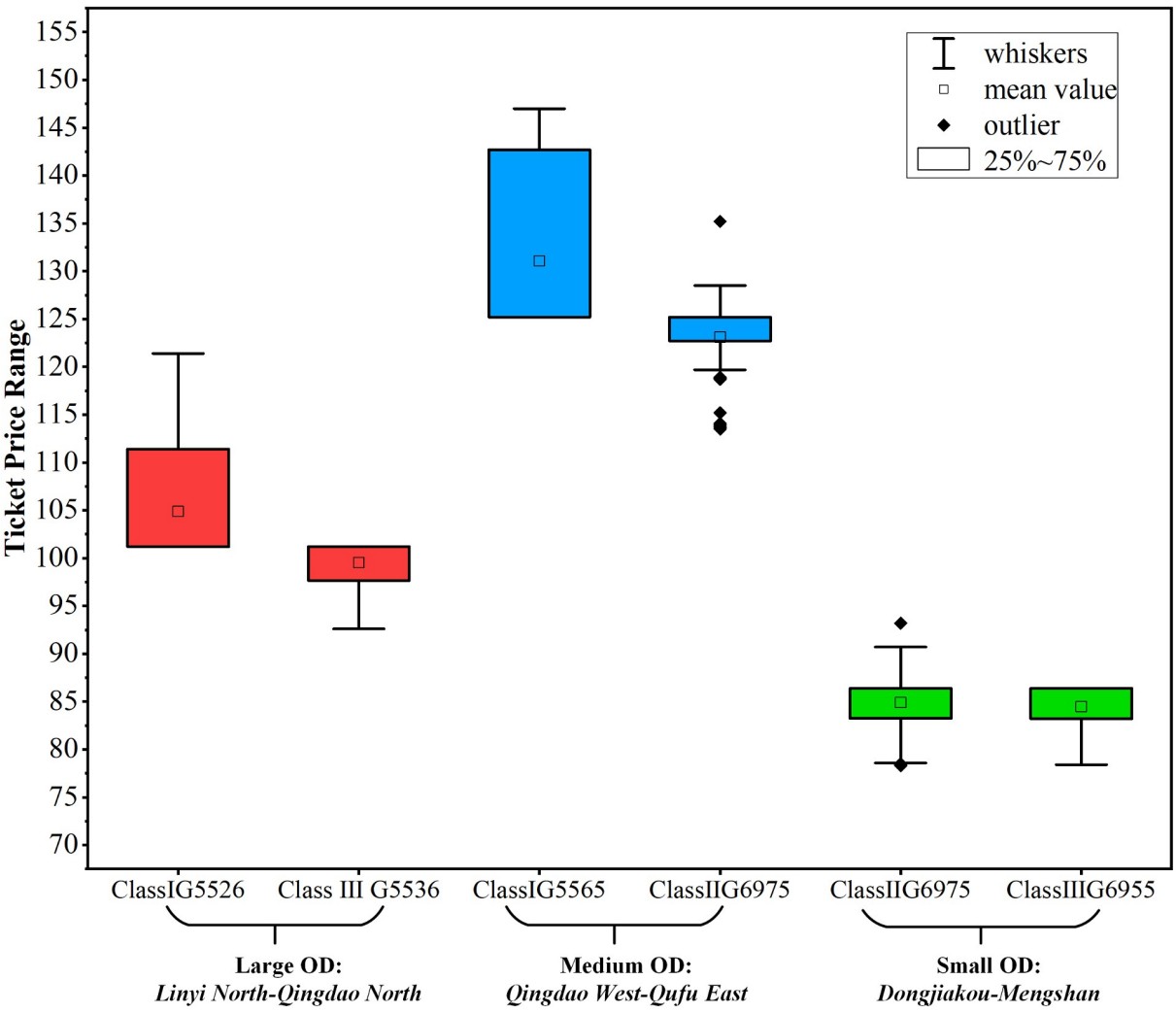

**Fig 9. Ticket price fluctuation for different classes of trains under different OD classes.**

of whiskers, *G5526* has only upper whisker and *G5536* has only lower whisker. The reason is that Class I only consider ticket price rate increase while Class III only consider ticket price rate decrease Table 8.

The ticket price fluctuation of different trains(Class I *G5565* and Class II*G6975*) under medium OD (*Qingdao West-Qufu East*) is calculated, as shown in the middle of Fig 9. In terms of ticket price dynamic range, the box range of *G5565* is wider than that of *G6975*, matching the basic situation that the ticket price fluctuation range of Class I is larger than that of ClassII. In terms of MEAN value, the average ticket price of *G5565* is higher than that of *G6975*. The reason why the MEAN value of *G6975* close to the box is that some outliers are small, which lowers the average ticket price. Generaly speaking, as long as the MEAN value is within two whiskers, the result can be reasonably analyzed. In terms of whiskers, *G5556* lacks lower whisker, which is consistent with the ticket price adjustment range in Table 8.

The ticket price fluctuation of different trains(ClassII *G6975* and Class III *G6955*) under small OD (*Dongjiakou-Mengshan*) is calculated, as shown on the right of Fig 9. In terms of

price dynamic range, the box range of *G6975* and *G6955* is similar. In terms of MEAN value, the average ticket price of *G6975* is higher than that of *G6955*. In terms of whiskers, *G6955* lacks upper whisker, aligning with the ticket price adjustment range in Table 8.

## 6.5 Algorithm core parameter sensitivity analysis

In the *LNS* algorithm, the destroy and repair parameters are key parameters that together define the neighborhood structure of *LNS*. The destroy method increases the diversity of the search space by breaking the current solution, while the repair method attempts to find a better solution within the broken solution. Generally speaking, the higher the ratio of destroy and repair operations, the more trains change price rate in each cycle.

In large-scale neighborhood search, if the parameter setting is too small, only a small part of the solutions is destroyed and repaired, then the heuristic search is difficult to search the whole neighborhood. If the parameter setting is too large, most of the solutions are broken, then the heuristic algorithm almost degenerates into repeated re-optimization.

In the scenario where the price adjustment range is set as [-20%,20%], destroy and repair parameters are set at 10% intervals. The average income value and calculation solving time curve are shown in Fig 10.

In the context of average revenue, a steady increase is observed as the parameter ratio escalates. With respect to calculation time, an exponential growth is evident as the parameter ratio is augmented.

Particularly, upon incrementing the parameter ratio from 50% to 60%, there is a modest acceleration in calculation time, concomitant with a marginal decline in average revenue. This observation underscores the significance of the 60% threshold as a critical reference point.

As the scale of data intensifies, it becomes imperative for railway staff to achieve a delicate equilibrium between optimizing total revenue and managing calculation time.

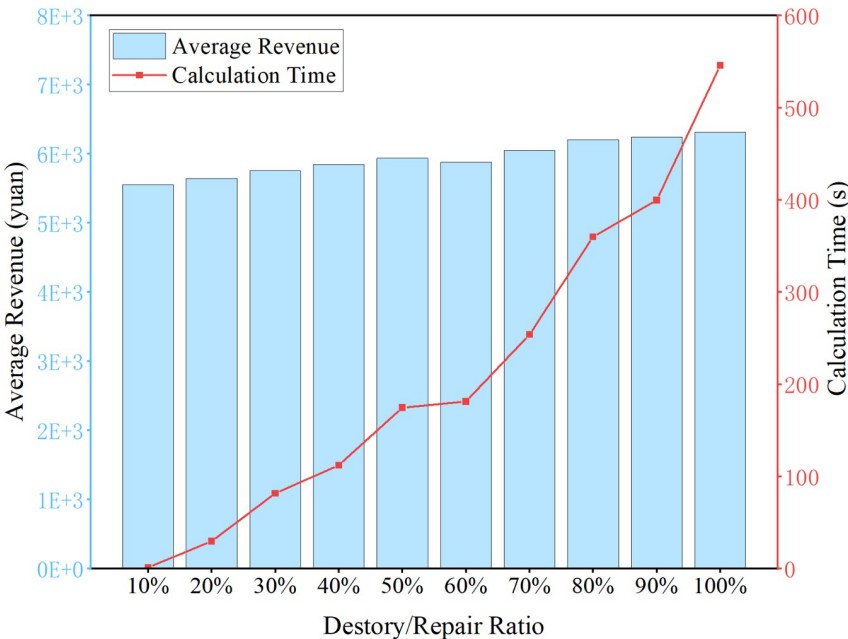

**Fig 10. Revenue comparison under different heuristic algorithm parameters.**

## 7 Conclusion

Aiming at the dynamic pricing of high-speed railway, this paper combines the revenue management concept with the differentiated products, and puts forward a train classification-based ticket price adjustment idea. Through the construction of the space-time service network, a multi-objective dynamic pricing model and a hybrid solution algorithm based on passenger flow assignment are proposed. The real operation data is used to make a case of *Shandong circular high-speed railway*. The total revenue under different price adjustment ranges and the ticket price for different classes of train under different levels ODs are analyzed, which prove the feasibility of the model and algorithm. Considering that the above idea of ticket price adjustment is easy to be applied in practice, it has greater operability in actual operation.

This paper studies the ticket price adjustment strategy of high-speed railway from the dimensions of station and train. Future research can be conducted from the following aspects:

1. Refine ticket price adjustment strategies at station level and train level. According to different scenarios, different station classification and train classification methods can be proposed. For the same seat class, the factors (such as historical passenger flow, historical turnover, etc.) can be combined. For the different seat classes, different ticket price adjustment ranges can be set.

2. Expand ticket price adjustment strategies at the pre-sale period and OD level. In the dimension of pre-sale period, the ticket price adjustment strategy can be studied on a daily basis or a multi-hour basis and adapted to real-time changes in passenger demand. In the OD level, different OD ticket price rates of the same train may fluctuate differently.

## Supporting information

**S1 Table. Train timetable data.**
(XLSX)

**S2 Table. Passenger flow data.**
(XLSX)

## Author Contributions

**Data curation:** Jiren CAO, Zhenhuan HE, Zhangjiaxuan LIU.

**Formal analysis:** Jiren CAO.

**Funding acquisition:** Lei NIE, Lu TONG.

**Investigation:** Zhangjiaxuan LIU.

**Methodology:** Jiren CAO, Zhenhuan HE.

**Project administration:** Lei NIE, Lu TONG.

**Resources:** Lu TONG.

**Software:** Jiren CAO, Zhenhuan HE.

**Supervision:** Lu TONG.

**Validation:** Lu TONG.

**Visualization:** Jiren CAO.

**Writing – original draft:** Jiren CAO.

**Writing – review & editing:** Jiren CAO.

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
