## [Decision Letter · Decision Letter 0]

30 Jul 2024

PONE-D-24-26027Dynamic Pricing Optimization for High-Speed Railway Based on Passenger Flow AssignmentPLOS ONE

Dear Dr. Tong,

Thank you for submitting your manuscript to PLOS ONE. After careful consideration, we feel that it has merit but does not fully meet PLOS ONE’s publication criteria as it currently stands. Therefore, we invite you to submit a revised version of the manuscript that addresses the points raised during the review process.

We look forward to receiving your revised manuscript.

Kind regards,

Levent ÇALLI, Ph.D

Academic Editor

PLOS ONE

Journal Requirements:

4. In the online submission form, you indicated that Data cannot be shared publicly because the authors do not have access to share the data. Data are available from the School of Transportation, Beijing Jiaotong University (contact the first author or corresponding author) for researchers who meet the criteria for access to confidential data.

Reviewers' comments:

Reviewer's Responses to Questions

**Comments to the Author**

1. Is the manuscript technically sound, and do the data support the conclusions?

Reviewer #1: Yes

Reviewer #2: Yes

2. Has the statistical analysis been performed appropriately and rigorously? 

Reviewer #1: Yes

Reviewer #2: Yes

3. Have the authors made all data underlying the findings in their manuscript fully available?

Reviewer #1: Yes

Reviewer #2: Yes

4. Is the manuscript presented in an intelligible fashion and written in standard English?

Reviewer #1: Yes

Reviewer #2: Yes

5. Review Comments to the Author

**Reviewer #1:** This paper combines the revenue management concept with the differentiated products, and puts forward a ticket price adjustment idea for dynamic pricing of high-speed railway. Through the construction of the space-time service network, a multi-objective dynamic pricing model and a hybrid solution algorithm based on passenger flow assignment are proposed. The case study proves the feasibility of the model and algorithm.

Some suggestions:

1) To sort out existing literature research using tables.

2) Labor ratio should be less than 0.7.

3) To give the dimension of M (=15).

4) To correct text errors: p6, "be chose"; p8," constraint, , ";p9,"、".

**Reviewer #2: **This paper studied pricing problem for HSR aiming to maximize railway revenue and passenger travel benefit. The authors established a multi-objective optimization model and adopted large neighborhood search heuristic algorithm to solve it. This research is interesting. However, there are some drawbacks needing to be improved. The major comments are given below:

1. English writing needs improvement.

2. The existing literature about passenger choice behavior and passenger flow assignment in this study are mainly related to public transportation, urban railway transit, dockless bike sharing systems. It is suggested to review related literature in railway.

3. The summary of the existing literature cannot reflect the research gaps between them and this study. The advantages of this paper are not clear. It is suggested to add a table to compare the existing studies with this work.

4. The ticket price of OD w to seat s in route k is the decision variable in this paper. The authors mentioned that the trains are decomposed into different classes. However, the subscript index of the decision variable does not include the train. It seems contradictory. In the numerical studies, the authors mentioned that only second-class seats are considered in case due to the availability of data. Thus, s is equal to 1 and the subscript s (seat s) does not seem to make sense.

5. The clarity of each figure in this paper needs to be enhanced.

6. It is “A to E” rather than “A to B” on line 194.

7. There are 5 nodes in Fig. 3 and 4 four nodes in Fig. 4. Is there any correspondence between Fig. 3 and Fig. 4?

8. The passenger time value is set as 50 yuan per person per hour in this study (line 238). How is it determined? What is the source of this value?

9. Equation 14 represents price not inverted constraint in a simple example. Please provide a general expression of the price not inverted constraint in a general situation.

10. The second objective function aims to maximize passenger travel benefit and solve the passenger assignment. Does Z2 consider the relationship between demand and the number of passengers? For example, the number of passengers should be smaller than the demand. How is the relationship between demand and assigned passenger flow reflected in this paper?

11. Equation 18 represents the transfer times constraint? It is suggested to provide the calculation formula of the number of transfer times.

12. In section 5.3, it is suggested to remove the step 2 in the solution steps.

13. How is the LNS algorithm designed (Step 3)? The clear algorithm steps and solving ideas should be given.

14. In step 4, the number of passengers on board appears to be allocated rather than optimized by the second objective function.

15. “high-speed” of the legend in Fig. 7 should be replaced by “HSR”.

16. Train classification is achieved according to the station levels rather than train attributes (or service level) such as speed, departure time, price and so on. What are the advantages of the classification method in this paper compared to the classification method according to train service level. Why do authors choose this classification method?

17. In the numerical studies, the revenue should be compared between two cases. Case 1 represents the train classification. Case 2 represents no train classification with the uniform price upper and lower bounds. This comparison could help reveal the meaning of this train classification method.

18. How is the subdivide range in Table 5 determined?

19. It is suggested to present the optimal results, such as the ticket price for each seat class between each OD served by each train.

20. In section 6.3, the price adjustment ranges have different upper bounds and uniform lower bounds. It is suggested to add different lower bounds to reflect the optimization rules more comprehensively.

21. It is suggested that the future research direction should be more detailed and specific.

6. PLOS authors have the option to publish the peer review history of their article (what does this mean?). If published, this will include your full peer review and any attached files.

Reviewer #1: No

Reviewer #2: No

---

## [Author Response · Author response to Decision Letter 0]

13 Sep 2024

Dear editor and reviewers,

On behalf of my co-authors, we thank you very much for giving us an opportunity to revise our manuscript. We appreciate editor and reviewers very much for their positive and constructive comments and suggestions on our manuscript entitled "Dynamic Pricing Optimization for High-Speed Railway Based on Passenger Flow Assignment" (PONE-D-24-26027).

These comments are very valuable and helpful for the improvement of our paper, and also have important guiding significance for our research. We have studied comments carefully and have made correction which we hope meet with approval.

We are very sorry to update the revised manuscript so late because of the case study redesign, calculation and analysis. 

We have noticed that the "Financial Disclosure" section of the newly generated PDF does not seem to be able to be modified. I need to explain to you that we have updated the fund information in both the system and the new cover letter.

Best wishes,

CAO Jiren

---

## [Decision Letter · Decision Letter 1]

9 Oct 2024

PONE-D-24-26027R1Dynamic pricing optimization for high-speed railway based on passenger flow assignmentPLOS ONE

Dear Dr. Tong,

Thank you for submitting your manuscript to PLOS ONE. After careful consideration, we feel that it has merit but does not fully meet PLOS ONE’s publication criteria as it currently stands. Therefore, we invite you to submit a revised version of the manuscript that addresses the points raised during the review process.

**ACADEMIC EDITOR: **Please upload the revised version in a clear format with "Track Changes" turned off. You need to highlight the changes made in the new manuscript. Otherwise, the document becomes complicated and very difficult to evaluate. 

We look forward to receiving your revised manuscript.

Kind regards,

Levent ÇALLI, Ph.D

Academic Editor

PLOS ONE

Reviewers' comments:

Reviewer's Responses to Questions

**Comments to the Author**

1. If the authors have adequately addressed your comments raised in a previous round of review and you feel that this manuscript is now acceptable for publication, you may indicate that here to bypass the “Comments to the Author” section, enter your conflict of interest statement in the “Confidential to Editor” section, and submit your "Accept" recommendation.

Reviewer #1: All comments have been addressed

Reviewer #3: (No Response)

Reviewer #4: All comments have been addressed

2. Is the manuscript technically sound, and do the data support the conclusions?

Reviewer #1: Yes

Reviewer #3: Partly

Reviewer #4: Yes

3. Has the statistical analysis been performed appropriately and rigorously? 

Reviewer #1: Yes

Reviewer #3: N/A

Reviewer #4: Yes

4. Have the authors made all data underlying the findings in their manuscript fully available?

Reviewer #1: Yes

Reviewer #3: No

Reviewer #4: Yes

5. Is the manuscript presented in an intelligible fashion and written in standard English?

Reviewer #1: Yes

Reviewer #3: Yes

Reviewer #4: Yes

6. Review Comments to the Author

Reviewer #1: All comments have been responed appropriately. Experiments have been conducted rigorously. The conclusions are drawn appropriately.

Reviewer #3: Your study tackles an important issue in the transportation sector by proposing a dynamic pricing model that considers passenger flow assignment. The integration of differentiated train products and dynamic pricing strategies is particularly interesting and has the potential to contribute significantly to the field.

However, I have several concerns that need to be addressed before the manuscript can be considered for publication. Below are detailed technical questions and comments that highlight areas of ambiguity and require clarification.

1. Model Justification: Can you elaborate on the rationale behind choosing the specific algorithms for passenger flow assignment? What criteria were used to evaluate their effectiveness?

2. Dynamic Pricing Mechanism: How does the dynamic pricing mechanism adapt to real-time changes in passenger demand? Are there specific thresholds or triggers that initiate price adjustments?

3. Data Sources: What specific operational data were used in your case study? How was the data validated to ensure its reliability?

4. Algorithm Complexity: The manuscript mentions the use of a large-scale neighborhood search algorithm. Can you provide more details on its implementation and computational complexity?

5. Passenger Flow Assignment: How do you ensure that the passenger flow assignment accurately reflects real-world scenarios? Are there any assumptions made that could impact the results?

6. Sensitivity Analysis: Have you conducted any sensitivity analysis to understand how changes in key parameters affect the outcomes of your model?

7. Comparison with Existing Models: How does your proposed model compare with existing dynamic pricing models in terms of performance and applicability? Are there any benchmarks used for comparison?

8. Implementation Challenges: What are the potential challenges in implementing your proposed pricing strategy in a real-world setting? How do you suggest overcoming these challenges?

9. Passenger Behavior Insights: How does your model account for passenger behavior and preferences? Are there any behavioral assumptions that could influence the pricing outcomes?

10. Economic Implications: Can you discuss the broader economic implications of your pricing strategy on the high-speed railway market and its stakeholders?

11. Limitations of the Study: What are the limitations of your study, and how might they affect the generalizability of your findings?

12. Future Research Directions: What future research directions do you envision based on the findings of your study? Are there specific areas that require further exploration?

I look forward to the authors' revisions and responses to these comments.

Reviewer #4: (No Response)

7. PLOS authors have the option to publish the peer review history of their article (what does this mean?). If published, this will include your full peer review and any attached files.

Reviewer #1: No

Reviewer #3: No

Reviewer #4: No

---

## [Author Response · Author response to Decision Letter 1]

6 Nov 2024

Dear editor and reviewers,

On behalf of my co-authors, I extend our heartfelt gratitude for the opportunity to revise our submission. We are deeply appreciative of the insightful and constructive feedback provided on our manuscript, titled "Dynamic Pricing Optimization for High-Speed Railway Based on Passenger Flow Assignment" (PONE-D-24-26027R1).

Your comments have been invaluable in enhancing the quality of our work and have offered significant guidance for our ongoing research endeavors. We have meticulously reviewed each point and have made the necessary revisions, which we believe address the concerns raised. We hope that our amendments will be found satisfactory.

Note: In the response letter, the explanatory text is in blue, the revised content in the manuscript is in green. ‘L’ shows the index of lines in the ‘Revised Manuscript with Track Changes’.

Best wishes,

CAO Jiren

---

## [Decision Letter · Decision Letter 2]

15 Nov 2024

Dynamic pricing optimization for high-speed railway based on passenger flow assignment

PONE-D-24-26027R2

Dear Dr. Tong,

We’re pleased to inform you that your manuscript has been judged scientifically suitable for publication and will be formally accepted for publication once it meets all outstanding technical requirements.

Kind regards,

Levent ÇALLI, Ph.D

Academic Editor

PLOS ONE

Additional Editor Comments (optional):

Reviewers' comments:

Reviewer's Responses to Questions

**Comments to the Author**

1. If the authors have adequately addressed your comments raised in a previous round of review and you feel that this manuscript is now acceptable for publication, you may indicate that here to bypass the “Comments to the Author” section, enter your conflict of interest statement in the “Confidential to Editor” section, and submit your "Accept" recommendation.

Reviewer #1: All comments have been addressed

Reviewer #3: All comments have been addressed

Reviewer #4: All comments have been addressed

2. Is the manuscript technically sound, and do the data support the conclusions?

Reviewer #1: Yes

Reviewer #3: Yes

Reviewer #4: Yes

3. Has the statistical analysis been performed appropriately and rigorously? 

Reviewer #1: Yes

Reviewer #3: N/A

Reviewer #4: Yes

4. Have the authors made all data underlying the findings in their manuscript fully available?

Reviewer #1: Yes

Reviewer #3: Yes

Reviewer #4: Yes

5. Is the manuscript presented in an intelligible fashion and written in standard English?

Reviewer #1: Yes

Reviewer #3: Yes

Reviewer #4: Yes

6. Review Comments to the Author

Reviewer #1: The authors have adequately addressed my comments. The statistical analysis has been performed appropriately.

Reviewer #3: All comments have been applied. Therefore, the manuscript has been improved and is ready for publication.

Reviewer #4: (No Response)

7. PLOS authors have the option to publish the peer review history of their article (what does this mean?). If published, this will include your full peer review and any attached files.

Reviewer #1: No

Reviewer #3: No

Reviewer #4: No

---

## [Editor Report · Acceptance letter]

20 Nov 2024

PONE-D-24-26027R2 

PLOS ONE

Dear Dr. Tong, 

I'm pleased to inform you that your manuscript has been deemed suitable for publication in PLOS ONE. Congratulations! Your manuscript is now being handed over to our production team.

Kind regards, 

on behalf of

Assoc. Prof. Levent ÇALLI 

Academic Editor

PLOS ONE